# The relationship between perceptions of electronic health record usability and clinical importance of social and environmental determinants of health on provider documentation

**Natasha Sood[1], Christy Stetter[2], Allen Kunselman[2], Sona Jasani[3]***

**1** Pennsylvania State College of Medicine, Hershey, Pennsylvania, United States of America, **2** Department of Public Health Sciences, Pennsylvania State College of Medicine, Hershey, Pennsylvania, United States of America, **3** Department of Obstetrics, Gynecology and Reproductive Sciences, Yale School of Medicine, New Haven, Connecticut, United States of America

* sona.jasani@yale.edu

**Data Availability Statement:** The data used in this submission was collected via a secure REDCap survey at Penn State College of Medicine and Penn

## Abstract

Social and environmental determinants of health (SEDH) data in the electronic health record (EHR) can be inaccurate and incomplete. Providers are in a unique position to impact this issue as they both obtain and enter this data, however, the variability in screening and documentation practices currently limits the ability to mobilize SEDH data for secondary uses. This study explores whether providers' perceptions of clinical importance of SEDH or EHR usability influenced data entry by analyzing two relationships: (1) provider charting behavior and clinical consideration of SEDH and (2) provider charting behavior and ease of EHR use in charting. We performed a cross-sectional study using an 11-question electronic survey to assess self-reported practices related to clinical consideration of SEDH elements, EHR usability and SEDH documentation of all staff physicians, identified using administrative list-serves, at Penn State Health Hershey Medical Center during September to October 2021. A total of 201 physicians responded to and completed the survey out of a possible 2,478 identified staff physicians (8.1% response rate). A five-point Likert scale from "never" to "always" assessed charting behavior and clinical consideration. Responses were dichotomized as consistent/inconsistent and vital/not vital respectively. EHR usability was assessed as "yes" or "no" responses. Fisher's exact tests assessed the relationship between charting behavior and clinical consideration and to compare charting practices between different SEDHs. Cumulative measures were constructed for consistent charting and ease of charting. A generalized linear mixed model (GLMM) compared SDH and EDH with respect to each cumulative measure and was quantified using odds ratios (OR) and 95% confidence intervals (CI). Our results show that provider documentation frequency of an SEDH is associated with perceived clinical utility as well as ease of charting and that providers were more likely to consistently chart on SDH versus EDH. Nuances in these relationships did exist with one notable example comparing the results of smoking (SDH) to infectious disease outbreaks (EDH). Despite similar percentages of physicians reporting that both smoking and infectious

State Health. The authors have provided access to the raw CSV file of the data along with supporting documentation in the supplementary information S2 and S3 Appendices.

**Funding:** The author(s) received no specific funding for this work.

**Competing interests:** The authors have declared that no competing interests exist.

disease outbreaks are vital to care, differences in charting consistency and ease of charting between these two were seen. Taken as a whole, our results suggest that SEDH quality optimization efforts cannot consider physician perceptions and EHR usability as siloed entities and that EHR design should not be the only target for intervention. The associations found in this study provide a starting point to understand the complexity in how clinical utility and EHR usability influence charting consistency of each SEDH element, however, further research is needed to understand how these relationships intersect at various levels in the SEDH data optimization process.

## Author summary

The widespread use of the electronic health record (EHR) in 2009 has accelerated the use of technology in clinical care. However, the current infrastructure of the EHR poses several barriers to effectively accommodate social and environmental determinants of health related to a patient's clinical care. Here, we sought to understand the relationship between provider perceptions of clinical importance of various social and environmental determinants of health as well as ease of charting these in the EHR on frequency of documentation of these determinants. We found that while participants reported that social and environmental determinants of health were important for their patient's care, this did not always influence their documentation practices of those determinants. Less than half of participants stated that documentation of determinants was easy however, certain determinants seemed to be "easier" than others. Additionally, the likelihood of documenting a specific determinant of health may be influenced by whether related determinants are also documented by a provider. Our study provides preliminary evidence that documentation frequency may depend on physician perceptions of health determinant integration into care delivery as well as EHR usability.

## Introduction

Systematic screening and documentation of health determinants can directly impact patient care [1–8] and research efforts [1,5]. Mobilizing this data from the electronic health record (EHR) is hindered by heterogeneous provider documentation practices and interoperability issues [1,2,9–22]. Data acquisition is also deficient with only 15.6% of physician practices and 24.4% of hospitals performing comprehensive health determinants screening per The National Survey of Healthcare Organizations and Systems Responses (2017–2018) [13]. Determinants of health include settings in which people are born, live, learn, work and age [23]. The Healthy People 2030 Initiative and The County Health Rankings Model, provide organizational schemas for health determinants [1,2,12]. These health frameworks help provide guidance on what data to capture but limitations do exist. For example, misclassification biases can result from using community level data due to limited stored environmental data points in health system EHRs. Lacking is an established standardized set of determinants or method of capture [5,13,14,19] to inform care delivery and optimization efforts.

Providers are in a unique position to influence the quality of determinants data within the EHR as they both acquire necessary elements from screening and record information [24,25]. Provider engagement (documenting, reviewing and updating information) with EHR health determinants data is associated with reduced hospital readmission rates; unfortunately, these

provider behaviors appear reduced for health determinants as compared to other medical history components [26]. Data quality relies on accuracy and completeness [27] and issues including inaccuracy and incompleteness have been identified with determinants of health data [28], consistent with the notion that the usability of EHR data is generally poor [29,30]. Structured data entry can facilitate data use and exchange [13] but providers often use non-structured formats to record health determinants information [1,14] which may be more difficult to extract for secondary use. Structured data entry may negatively impact clinical workflows [31] and negative EHR user experiences in general may result in partial data capture [29,32,33]. Interestingly, provider collection and capture may be modifiable; those who care for vulnerable populations or routinely use social care services are more likely to engage with EHR determinants data [34]. Whether data quality issues are related to incomplete entry, incomplete capture or a lack of universal guidelines is unclear from the existing literature.

The social elements of health determinants include housing, food security, transportation, utilities, childcare, stable employment, education, financial stability, personal safety, access to guns in the household, insurance, smoking and refugee status [14]. For the purposes of this study, these above elements will be considered as social determinants of health (SDH). Given the amassing evidence of environmental influencers on wellness and disease [9,35] an expanded set of environmental elements of health determinants will be used and include air pollution, household pollution and toxicities, extreme weather, access to basic appliances, exposure to natural disasters, infectious disease outbreaks and proximity to landfills/industrial centers/waste processing plants. For the purposes of this study, these above elements will be considered as environmental determinants of health (EDH) and were compiled based on Healthy People 2030 [23], the World Health Organization and the Pan American Health Organization [36]. Together, SDH and EDH will be considered under the term social and environmental determinants of health (SEDH) for this study. SEDH data quality and mobilization depends on collection and storage, two tasks performed by providers whose engagement with SEDH data may be modifiable. Our team was therefore interested in exploring whether providers' perceptions of clinical importance or EHR usability influenced data entry. The objective of this study was to identify two possible relationships at an academic medical center: (1) the relationship between provider self-reported charting behavior and self-reported clinical consideration of SEDH and (2) the relationship between provider self-reported charting behavior and self-reported ease of EHR use in charting SEDH. We relied on self-reported charting behavior because routine structured data entry of SEDH elements was limited at the study institution and because we did not want to exclude any possible provider SEDH documentation modality. We hypothesize that for each individual SEDH element, the frequency of charting would be positively correlated with the frequency of clinical consideration as well as the ease of EHR use. We analyzed our data by looking at each individual SEDH element, accounted for an expanded set of EDHs, and analyzed for documentation behaviors related to groups of determinants.

## Materials and methods

### Study design

We performed a cross-sectional study using an electronic survey to assess self-reported practices related to clinical consideration of SEDH elements, EHR usability and SEDH documentation of all staff physicians at Penn State Health Hershey Medical Center during September to October 2021. Staff physicians were identified using administrative listservs in the Medical Staff and Graduate Medical Education Offices. Non-study team administrative personnel in

these offices distributed the survey electronically using staff physician institutional email addresses via these administrative listservs to ensure anonymous participation.

### Data collection tool

REDCap was used to create an electronic survey that consisted of 11-questions (S1 Appendix). The first five items obtained basic demographic data including participant role, practice setting, department/medical specialty, gender and ethnicity. The next three items asked participants questions regarding charting behavior, frequency of routine clinical consideration and EHR usability for charting each SDH element. The last three items asked participants questions regarding charting behavior, frequency of routine clinical consideration and EHR usability for charting each EDH element. A five-point Likert scale from "never" to "always" was used to assess participant charting behavior and clinical consideration as specifically related to the participant's own clinical practice. EHR usability was assessed by asking participants to answer "yes" or "no" to whether the EHR makes documenting each SEDH element easy.

### Statistical analysis

Charting behavior was dichotomized as consistent (often/always) and inconsistent (never, rarely, sometimes), and clinical consideration was dichotomized as vital (often/always) and not vital (never/rarely/sometimes). Fisher's exact tests were used to assess the relationship between charting behavior and clinical consideration of each SEDH, as well as to compare charting practices between different SEDHs. As a cumulative measure, physicians were categorized as consistently charting on at least 1 SDH (yes/no) and at least 1 EDH (yes/no). A similar cumulative measure was constructed for easy to chart on at least 1 SDH/EDH. A generalized linear mixed model (GLMM) was used to compare the two types of determinants of health (SDH vs. EDH) with respect to each cumulative measure. The GLMM used a binary distribution with a logit link and consisted of a fixed type effect (SDH vs. EDH) and random physician effect to account for correlation between the SDH and EDH responses per physician. The results from the GLMM were quantified using odds ratios (OR) and 95% confidence intervals (CI). Statistical analyses were performed using SAS software, version 9.4 (SAS Institute Inc., Cary, NC).

### Ethics statement

The Institutional Review Board (IRB) at the study institution granted exempt status for this research (#STUDY00018033). Approval was obtained prior to conducting the study and no incentives were offered for participation. The Human Subjects Protection Officer at the study institution determined that the study did not require full IRB review because the research met criteria for exempt research according to the policies of the study institutions and the provisions of applicable federal regulations. The first page of the survey contained a summary explanation of research which outlined that completing the survey indicated consent to participate in the study. Participation was voluntary and subjects could choose to withdraw or close the survey at any time. No identifiable information was collected as the survey was anonymous.

## Results

### Baseline characteristics

A total of 2,478 staff physicians (1,800 attending physicians and 678 residents/fellows) were identified using an administrative listserv at the study institution during the project timeframe. Of this total, 201 participants completed the survey (8.1% response rate) and were included in

**Table 1. Demographics.**

| Survey Question | Frequency | Percent* |
|---|---|---|
| **Role in the health system** | | |
| Resident | 24 | 11.9 |
| Fellow | 6 | 3.0 |
| Attending | 168 | 83.6 |
| Other | 3 | 1.5 |
| **Setting of Practice** | | |
| Inpatient (only) | 21 | 10.4 |
| Outpatient (only) | 54 | 26.9 |
| Both inpatient and outpatient | 126 | 62.7 |
| **Department** | | |
| Anesthesiology | 6 | 3.0 |
| Dermatology | 3 | 1.5 |
| Family Medicine | 24 | 11.9 |
| Internal Medicine & Subspecialties | 40 | 19.9 |
| Neurology | 5 | 2.5 |
| Obstetrics & Gynecology | 11 | 5.5 |
| Pediatrics | 51 | 25.4 |
| Psychiatry | 11 | 5.5 |
| Surgery & Surgical Subspecialties | 27 | 13.4 |
| Emergency Medicine | 6 | 3.0 |
| Pathology | 1 | 0.5 |
| Physical Medicine & Rehabilitation | 4 | 2.0 |
| Radiology | 6 | 3.0 |
| Other | 6 | 3.0 |
| **Gender** | | |
| Male | 96 | 47.8 |
| Female | 97 | 48.3 |
| Prefer not to answer | 8 | 4.0 |
| **Ethnicity** | | |
| Hispanic/Latino | 11 | 5.5 |
| Not Hispanic/Latino | 174 | 86.6 |
| Prefer not to answer | 16 | 8.0 |

* Due to rounding, percentages may sum >100%.

the analysis. Table 1 reports the baseline characteristics of these participants. The majority of physicians were attendings (84%) and most practiced both inpatient and outpatient medicine (63%). Approximately 63% were primary care providers and 37.1% were specialists; percentages from each department are available in Table 1. The majority of participants were not Hispanic/Latino and there were an equal number of male and female participants.

## SDH behaviors and perceptions

**SDH charting frequency.** Of the 13 SDHs assessed, smoking was the only SDH that was consistently charted on by at least half of physicians (78%) with all other SDHs being consistently charted on by less than half (Fig 1). Dermatology (33%) and Radiology (17%) were the only two departments that had less than half of their physicians charting on smoking and

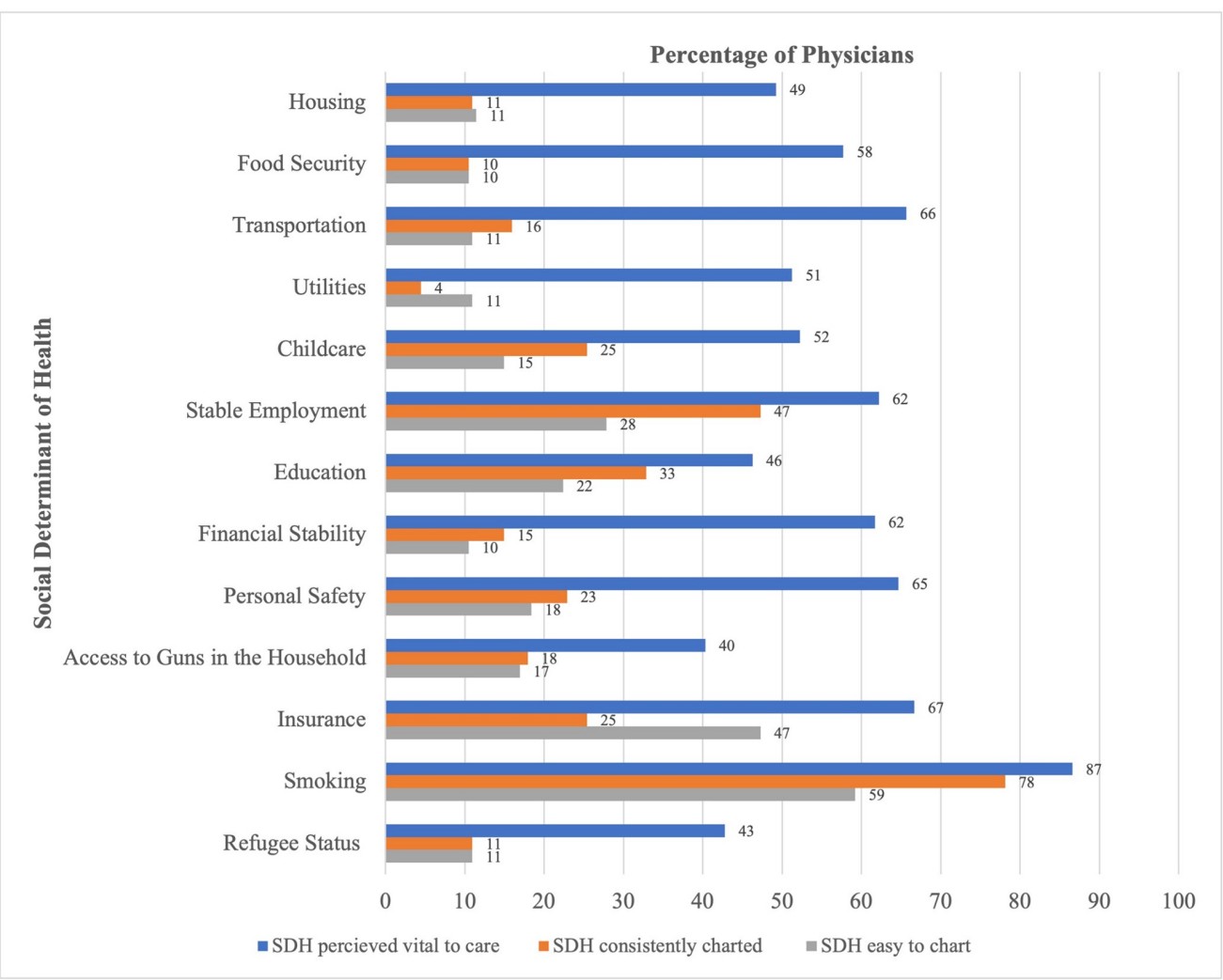

**Fig 1. Summary of SDH perceptions regarding clinical consideration (vital to care), consistent charting, and ease of charting.** Fig 1 demonstrates the percentage of physicians who included social determinants of health (SDH) in their clinical considerations (i.e. if they indicated that SDHs were vital to their care) and the percentage of physicians who consistently charted on SDHs. The figure also demonstrates the percentage of physicians who indicated that the SDH was easy to chart.

Psychiatry was the only department who did not report smoking as their top charted SDH (S1 Table).

**Perceived importance of SDH.** Of the 13 SDHs assessed, 9 were reported as vital to their patient care by at least half of physicians with the SDH of smoking reported by the highest percentage of physicians (87%) (Fig 1.). The SDHs that were reported by less than half were education (46%), refugee status (43%), housing (49%) and access to guns in the household (40%) (Fig 1.). Though a majority of departments reported smoking to be a top SDH that was vital to care, there was more variety in the top SDHs being vital to care, by department, as compared to the top charted SDH (S1 Table).

**Ease of charting SDH.** In general, respondents reported that SDHs were difficult to chart in the EHR. Smoking and insurance were reported as easy to chart by 59% and 47% of physicians respectively; rates for the other 11 SDHs were less than 30% (Fig 1).

**Table 2. Charting of SDH factors based on perceived importance and ease of charting in EHR.**

| SDH | Perceived Vital | | | Easy to Chart | | |
|---|---|---|---|---|---|---|
| | Yes | No | P-value | Yes | No | P-value |
| | n (%) charted consistently | n (%) charted consistently | | n (%) charted consistently | n (%) charted consistently | |
| Housing | 18 (18.2) | 4 (3.9) | 0.001 | 4 (17.4) | 18 (10.1) | 0.29 |
| Food security | 19 (16.4) | 2 (2.4) | < .001 | 5 (23.8) | 16 (8.9) | 0.05 |
| Transportation | 30 (22.7) | 2 (2.9) | < .001 | 6 (27.3) | 26 (14.5) | 0.13 |
| Utilities | 8 (7.8) | 1 (1.0) | 0.04 | 2 (9.1) | 7 (3.9) | 0.26 |
| Childcare | 44 (41.9) | 7 (7.3) | < .001 | 17 (56.7) | 34 (19.9) | < .001 |
| Employment | 73 (58.4) | 22 (28.9) | < .001 | 37 (66.1) | 58 (40.0) | 0.001 |
| Education | 44 (47.3) | 22 (20.4) | < .001 | 20 (44.4) | 46 (29.5) | 0.07 |
| Financial stability | 28 (22.6) | 2 (2.6) | < .001 | 4 (19.0) | 26 (14.4) | 0.53 |
| Personal safety | 43 (33.1) | 3 (4.2) | < .001 | 12 (32.4) | 34 (20.7) | 0.13 |
| Access to guns in the household | 30 (37.0) | 6 (5.0) | < .001 | 10 (29.4) | 26 (15.6) | 0.08 |
| Insurance | 44 (32.8) | 7 (10.4) | < .001 | 36 (37.9) | 15 (14.2) | < .001 |
| Smoking | 151 (86.8) | 6 (22.2) | < .001 | 105 (88.2) | 52 (63.4) | < .001 |
| Refugee status | 19 (22.1) | 3 (2.6) | < .001 | 6 (27.3) | 16 (8.9) | 0.02 |

**SDH charting consistency association with either perceived clinical importance or ease of charting.** Table 2 demonstrates the relationship of consistent charting with either the perception of clinical utility of that SDH (vital to practice) or the perception of how easy that SDH was to chart among all 13 SDHs assessed. Physician perception of whether an SDH was vital to patient care was significantly associated with consistent charting of that SDH for all SDHs assess. Physician perception of the ease of charting of an SDH, however, was significantly associated with consistent charting of the SDHs of childcare, employment, insurance, smoking, and refugee status only.

## EDH behaviors and perceptions

**EDH charting frequency.** Of the 7 EDHs assessed in the survey, the EDH of infectious disease outbreaks was consistently charted on by the most physicians (31%) (Fig 2). This, however, was far less than that for the SDH of smoking (78%). Most departments had less than half of their physicians charting on the EDH of infectious disease outbreaks with the exception of Pathology (100%), Psychiatry (55%), and Emergency Medicine (50%) (S2 Table). Less than 5% of physicians consistently charted on the other 6 EDHs (Fig 2).

**Perceived importance of EDH.** Of the 7 EDHs assessed, 3 were reported as vital to patient care by at least half of physicians (Fig 2). Infectious disease had the highest percentage of physicians who perceived this EDH as vital to patient care (80%) followed by access to basic appliances (59%) and household pollution (59%). Infectious disease outbreaks was a top EDH for both consistent charting and being vital to clinical practice (S2 Table).

**Ease of charting EDH.** Similar to SDHs, respondents reported that EDHs were difficult to chart in the EHR. The EDH of infectious disease outbreaks was reported as easy to chart by only 16.4% of physicians; rates for the other 6 EDHs were all less than 10% (Fig 2).

**EDH charting consistency association with either perceived clinical importance or ease of charting.** Table 3 demonstrates the relationship of consistent charting with either the perception of clinical utility of that EDH (vital to practice) or the perception of how easy that EDH was to chart among all 7 EDHs assessed. Physician perception of whether an EDH was

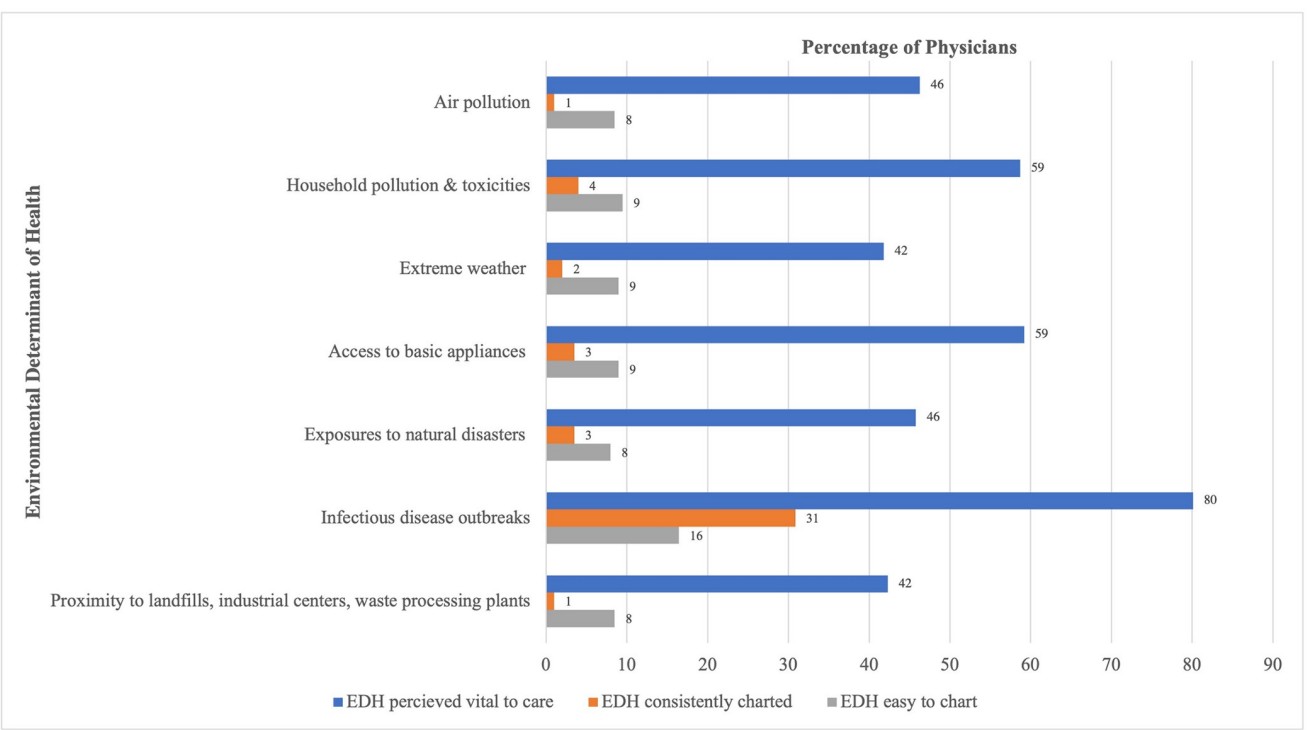

**Fig 2. Summary of EDH perceptions regarding clinical consideration (vital to care), consistent charting, and ease of charting.** Fig 2 demonstrates the percentage of physicians who included environmental determinants of health (EDH) in their clinical considerations (i.e. if they indicated that EDHs were vital to their care) and the percentage of physicians who consistently charted on EDHs. The figure also demonstrates the percentage of physicians who indicated that the EDH was easy to chart.

vital to patient care was significantly associated with consistent charting of the EDH for infectious disease outbreaks and access to basic appliances only. Physician perception of the ease of charting of an EDH was significantly associated with consistent charting of the EDH of infectious disease outbreaks only.

## General associations of SEDH charting practices

Several associations between SEDH charting practices were found. The percentage of physicians charting on food security (SDH) was higher for those who charted on transportation

**Table 3. Charting of EDH factors based on perceived importance and ease of charting in EHR.**

| EDH | Perceived Vital | | | Easy to Chart | | |
|---|---|---|---|---|---|---|
| | Yes | No | P-value | Yes | No | P-value |
| | n (%) charted consistently | n (%) charted consistently | | n (%) charted consistently | n (%) charted consistently | |
| Air pollution | 2 (2.2) | 0 (0.0) | 0.21 | 0 (0.0) | 2 (1.1) | 1.00 |
| Household pollution & toxicities | 7 (5.9) | 1 (1.2) | 0.14 | 2 (10.5) | 6 (3.3) | 0.17 |
| Extreme weather | 3 (3.6) | 1 (0.9) | 0.31 | 1 (5.6) | 3 (1.6) | 0.31 |
| Access to basic appliances | 7 (5.9) | 0 (0.0) | 0.04 | 2 (11.1) | 5 (2.7) | 0.12 |
| Exposure to natural disasters | 5 (5.4) | 2 (1.8) | 0.25 | 1 (6.3) | 6 (3.2) | 0.45 |
| Infectious disease | 61 (37.9) | 1 (2.5) | < .001 | 16 (48.5) | 46 (27.4) | 0.02 |
| Proximity to landfills, industrial centers, waste processing plants | 2 (2.4) | 0 (0.0) | 0.18 | 0 (0.0) | 2 (1.1) | 1.00 |

(SDH) compared to those who did not (34.4% vs. 5.9%, p < 0.001). The percentage charting on access to guns in the household (SDH) was higher for those who charted on personal safety (SDH) compared to those who did not (50.0% vs. 8.4%, p < 0.001). The percentage charting on refugee status (SDH) was higher for those who charted on exposure to natural disasters (EDH) compared to those who did not (100% vs. 7.7%, p < 0.001). Finally, the percentage charting on household pollution and toxicities (EDH) was higher for those who charted on housing (SDH) compared to those who did not (18.2% vs. 2.2%, p = 0.006).

A majority of physicians consistently charted on at least 1 SDH (88.1%), whereas fewer charted on at least 1 EDH (34.3%). The odds of consistently charting on at least 1 SDH were 14 times the odds of charting on at least 1 EDH (OR = 14.8, 95% CI (8.8, 24.9), p < .001). More than half (68.2%) of physicians reported that at least 1 SDH was easy to chart, compared to 18.4% of physicians who reported ease in charting at least 1 EDH. The odds of easily charting at least 1 SDH were 10 times the odds of easily charting at least 1 EDH (OR = 10.8, 95% CI (6.7, 17.5), p < .001).

## Discussion and conclusion

Generally, our data supports our hypothesis that frequency of charting of an SEDH is associated with physician perception of clinical importance or EHR usability, however, not every association measured in our study showed statistical significance. Regarding all SDHs, charting frequency was significantly associated with physician perception of clinical importance. Charting frequency was significantly associated with ease of charting for all SDHs except for housing, transportation, utilities, education, financial stability, personal safety, and access to guns. However, trends in the raw data for these SDHs do show that a higher percentage of physicians consistently charted SDHs if they perceived charting to be easy (Table 2). The lack of statistical significance could be likely due to the low overall numbers of respondents who reported that SDH charting was easy. With regards to EDH, charting frequency was significantly associated with physician perception of clinical importance for infectious disease outbreaks and access to basic appliances only. Charting frequency was significantly associated with ease of charting for the EDH of infectious disease outbreaks only. However, just as with SDH, trends in the raw EDH data do show that a higher percentage of physicians consistently charted on EDHs if they perceived the EDH to be vital or easy to chart (Table 3). Lack of statistical significance for EDH associations is likely due to the overall low numbers of those who reported consistently charting on EDH.

Overall, 43%-87% of physicians considered an SDH as vital to their patient care with 9 out of the 13 SDHs assessed being considered vital by over half of respondents. Unsurprising, smoking was considered to be the most vital SDH. Assessing for smoking status is deeply entrenched across all levels of medical training and therefore, the high rates of clinical consideration and consistent charting of this SDH was not surprising. What was notable was that the reported ease of charting for smoking was also higher than most other SDHs and EDHs (Fig 1). Systematized use of electronic screening forms, such as PRAPARE, was not a standardized practice at the study institution and provider SEDH documentation may vary between different physicians. Regardless of this possible variation (using structured data entry or an unstructured form like a clinical note) a physician would have been able to use their same chosen method to chart on all SEDHs. It is surprising, therefore, to see in our data a difference in the reported ease of charting for smoking as compared to all other SEDHs. Our data suggests that for the SDH of smoking, charting difficulty was less of an issue despite the fact that the process for charting on smoking would have been the same as for any other SDH or EDH for a particular provider. From our data, it is unclear if the perception of EHR usability for smoking

documentation was influenced by medical training of smoking screening. This does, however, pose an interesting question about whether downstream SEDH data quality issues can be modulated from upstream factors in the medical training of providers such as emphasizing SEDH screening [37] or actively integrating SEDH into care delivery [29].

As compared to SDH, a similar percentage of physicians (42–80%) considered an EDH as vital to their patient care with 3 out of 7 being considered vital by over half of respondents. Though the EDH of infectious disease outbreaks was considered to be the most vital to practice (80%) as compared to all other EDHs, the relatively low rates of consistent charting of this EDH (31%) was surprising given that our survey was conducted during the COVID-19 pandemic. Less than half of physicians in all departments (with the exception of Psychiatry and Pathology) reported charting on infectious disease outbreaks. It is possible that other members of the care team charted information about the patient's infectious disease status or exposure at various points of the patient's clinical care. However, the study team would have expected that the provider would have also documented this information either for screening or medical decision-making purposes. As with smoking, the slightly higher reported rates of ease of charting on this EDH suggests that EHR design may not be the only factor that influences physician documentation. Given the COVID-19 pandemic and reported high clinical consideration of infectious disease outbreaks, provider perceptions on EHR usability for documenting infectious disease outbreaks may have been influenced by the various practice changes dictated by the pandemic. Though our data cannot explain why we detected variations between the EDH of infectious disease as compared to the other EDHs, future studies should be undertaken to identify best practices regarding EDH consideration and documentation.

Our data also highlights some noteworthy associations and behavioral patterns with respect to SEDH data entry. Our results show that SEDH provider documentation is inconsistent: smoking was the only SDH that was consistently charted on by greater than half of physicians and no EDH was consistently charted on by greater than half. This finding is congruent with SEDH data quality issues noted in the literature [28–30]. Our data also shows discrepancies in provider considerations between SDH and EDH. The odds of physicians consistently charting on at least one SDH was 14 times the odds of consistently charting on at least one EDH. The odds of easily charting on at least one SDH were 10 times the odds of easily charting on at least one EDH. Despite using an expanded list of EDHs, the low rates of EDH charting or reported ease of charting was likely not due to reduced perceived clinical importance (Fig 1). Finally our results showed statistically significant associations in documenting behavior when grouping certain SEDH elements: (1) there was a higher likelihood to chart on food security if charting on transportation; (2) there was a higher likelihood to chart on access to guns if charting on personal safety; (3) there was a higher likelihood to chart on refugee status if charting on exposure to natural disasters; (4) there was a higher likelihood to chart on household pollution and toxicities if charting on housing. Considering these associations and the imbalance between SDH versus EDH charting practices may help guide interventions targeted towards improving SEDH data quality issues.

This study has several limitations. We had a low overall response rate of 8% which could lead to non-response bias. This low response resulted in our inability to stratify analyses by medical specialty. It is possible that the low response rate was influenced by the COVID-19 pandemic as physician activities were predominantly focused on patient care activities as opposed to research participation during the study period. Second, this study was conducted at a single academic medical center, possibly limiting the generalizability of the results. One advantage of the self-reported data in our study, however, was that we were able to assess for all possible types of charting behaviors and not only rely on discrete elements from structured data entry. Furthermore, physician responses seen in our study corroborated patterns seen in

the literature. Finally, we were unable to correlate physician responses to a health system level assessment of SEDH data quality or patient outcomes secondary to low structured documentation modalities and an inability to utilize natural language processing analytics for this particular study.

In conclusion, our study results show that provider documentation frequency of an SEDH is associated with clinical utility as well as ease of charting and that providers were more likely to consistently chart on SDH versus EDH. There is nuance, however, in these associations. Though a comparable percentage of physicians reported that smoking (SDH) and infectious disease outbreaks (EDH) were vital to care, there were still notable differences in charting consistency and ease of charting between these two. Inconsistent and incomplete acquisition affects the quality and use of SEDH data so understanding the factors that promote or hinder provider capture and entry should be a central consideration in any optimization effort. These efforts however cannot only focus on EHR design or physician perception as siloed entities, as suggested from our results. Instead, it may be more beneficial to approach data quality optimization using a multi-level framework: up-stream factors (examining education and training curricula), care delivery factors (establishing guidelines to systematize collection and storage) and down-stream factors (providing feedback through metrics on clinical use of SEDH elements and documentation practices). Our results provide a starting point to understand how clinical utility and EHR usability influence physician documentation practices for each individual SEDH element. Further research is still needed to explicate these associations at each of these three levels as well as to identify sustainable strategies to improve SEDH data quality and utility.

## Supporting information

**S1 Appendix. REDCap electronic survey.** First 5 survey items assess basic demographic data, next 3 items assess SDH information (charting behavior, frequency of routine clinical consideration, EHR usability) and last 3 items assess EDH information (charting behavior, frequency of routine clinical consideration, EHR usability).
(DOCX)

**S2 Appendix. REDCap electronic survey raw data file.** The data used in this submission was collected via a secure REDCap survey at Penn State College of Medicine and Penn State Health. The authors have provided access to the raw CSV file of the data in this supplementary information file.
(CSV)

**S3 Appendix. REDCap electronic survey supporting documentation data file.** The data used in this submission was collected via a secure REDCap survey at Penn State College of Medicine and Penn State Health. The authors have provided access to the supporting documentation in this supplementary information file.
(CSV)

**S1 Table. Top charted and clinically vital SDHs reported by department.** Table values correspond to the percentage of physicians by department who reported consistent charting on or vital to their practice. Only the SDHs with the highest percentage of physicians are listed. Multiple SDHs are listed if they had the same percentage of physicians reporting them.
(DOCX)

**S2 Table. Top charted and clinically vital EDHs reported by department.** Table values correspond to the percentage of physicians by department who reported consistent charting on or

vital to their practice. Only the EDHs with the highest percentage of physicians are listed. Multiple EDHs are listed if they had the same percentage of physicians reporting them. (DOCX)

## Acknowledgments

The study team would like to thank the administrative staff in the Penn State College of Medicine Department of Obstetrics and Gynecology, as well as the administrative staff in the Penn State Health Hershey Medical Center Graduate Medical Education Office and Medical Staff Office (Beth Herman and Eddie Keller), in helping to distribute the survey used in this study. The team also wishes to acknowledge the participants who provided their insights in this project.

## Author Contributions

**Conceptualization:** Natasha Sood, Sona Jasani.

**Data curation:** Natasha Sood, Sona Jasani.

**Formal analysis:** Christy Stetter, Allen Kunselman.

**Investigation:** Natasha Sood, Sona Jasani.

**Methodology:** Natasha Sood, Sona Jasani.

**Project administration:** Natasha Sood, Sona Jasani.

**Resources:** Natasha Sood, Sona Jasani.

**Supervision:** Sona Jasani.

**Validation:** Natasha Sood.

**Writing – original draft:** Natasha Sood, Sona Jasani.

**Writing – review & editing:** Natasha Sood, Christy Stetter, Sona Jasani.

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
