## [Decision Letter · Decision Letter 0]

15 Aug 2023

PDIG-D-23-00183

The relationship between perceptions of electronic health record usability and clinical importance of social and environmental determinants of health on provider documentation

PLOS Digital Health

Dear Dr. Sood,

Thank you for submitting your manuscript to PLOS Digital Health. After careful consideration, we feel that it has merit but does not fully meet PLOS Digital Health's publication criteria as it currently stands. Therefore, we invite you to submit a revised version of the manuscript that addresses the points raised during the review process.

Please submit your revised manuscript within 60 days Oct 14 2023 11:59PM. If you will need more time than this to complete your revisions, please reply to this message or contact the journal office at digitalhealth@plos.org. Please include the following items when submitting your revised manuscript:

We look forward to receiving your revised manuscript.

Kind regards,

Shlomo Berkovsky

Section Editor

PLOS Digital Health

Journal Requirements:

Additional Editor Comments (if provided):

Reviewers' comments:

Reviewer's Responses to Questions

**Comments to the Author**

1. Does this manuscript meet PLOS Digital Health’s publication criteria? Is the manuscript technically sound, and do the data support the conclusions? The manuscript must describe methodologically and ethically rigorous research with conclusions that are appropriately drawn based on the data presented.

Reviewer #1: Yes

Reviewer #2: Yes

Reviewer #3: Partly

2. Has the statistical analysis been performed appropriately and rigorously?

Reviewer #1: Yes

Reviewer #2: Yes

Reviewer #3: Yes

3. Have the authors made all data underlying the findings in their manuscript fully available (please refer to the Data Availability Statement at the start of the manuscript PDF file)?

Reviewer #1: Yes

Reviewer #2: No

Reviewer #3: Yes

4. Is the manuscript presented in an intelligible fashion and written in standard English?

Reviewer #1: Yes

Reviewer #2: Yes

Reviewer #3: No

5. Review Comments to the Author

Reviewer #1: Dear Editors, 

At first, while reading, and understanding the study conducted, I assessed whether the authors set out and completed what they agreed to do. And, without a doubt, the study is clearly written, and well thought through. While, the auhtors acknowledge their levels of uncertainty, and represent the findings well, one cannot fault them on the basis of a self-reported dataset. 

In future, it would be important to conduct a sub-class analysis of the physicians, and the differences that could potentially be reported by these physicians. Yet, as it stands, the authors provide a rigorous study and coherent argument for the findings in the time of COVID-19 pandemic. The findings account for the differences of whether providers’ perceptions of clinical

importance of SEDH or EHR usability influenced data entry by analyzing two

relationships: (1) provider charting behavior and clinical consideration of SEDH and (2)

provider charting behavior and ease of EHR use in charting. 

Furthermore, as the reviewer, I do acknowledge that this was an exploratory study. However, it would have been useful to include the baseline knowledge of the physicians in the study. Also, how this influence the provider's perceptions of the clinical importance of SEDH and EHR usability. Perhaps, for future purposes, the authors should consider baseline knowledge of the participants, as well as the frequency of entries and how this influence provider charting behaviour and ease of use EHR use in charting.

Reviewer #2: This is a competently written paper but only involves the input from physicians at one academic institution limiting its generalisability. The paper fails to add anything of significant originality to the literature. I would not recommend publication of this paper.

Reviewer #3: Comments to authors

General comments. Overall structure of the manuscript should include sub themes describing the processes carried out during the study. For example under material and methods section, data collection, sampling process and study design has been boxed together. There are component under the study design which should be moved to the introduction section to bring clarity to the stated acronyms

Introduction

What are some of the elements of SEDH? Can the authors be clear on what they define as SEDH for readers to appreciate what to expect? Also can the authors clarify what EDH is ? The introduction is not clear on how the providers come to be associated with SEDH and EDH. The manuscript can be improved if the authors state and clarify these issues early. As it stands now, the understanding and interpretation of SEDH and EDH are to the discretion of the reader.

Although the Objectives of the study are stated clearly, the lack of clarity of SEDH, EDH and how providers are associated with it at the onset, obscures how the stated objectives can be achieved. 

Methods

Under study design, this should state clearly the study type, when and where it was conducted. The text describing the data collection tools should be captured separately under different subtheme to bring clarity. Again, there should be a section on how data was collected, and the procedures involved to aid understanding how the study was carried out

Again the part of text under the survey design which defines SEDH, EDH SDH should be moved under the introduction

How were participants selected or recruited into the study? How was sampling and sample size determined?

Results

Line 173-174 the authors states: “ The survey was administered to all physicians at a single academic health center, of which a total of 201 completed the survey…” How many physicians were in this facility? What was the response rate? 

Discussion

Again on Line 353 the authors’ state “We had a low overall response rate of 8% which…” How did the authors arrived at this figure?

6. PLOS authors have the option to publish the peer review history of their article (what does this mean?). If published, this will include your full peer review and any attached files.

**Do you want your identity to be public for this peer review?** For information about this choice, including consent withdrawal, please see our Privacy Policy.

Reviewer #1: No

Reviewer #2: Yes: Dr Sarah Markham

Reviewer #3: No

---

## [Decision Letter · Decision Letter 1]

30 Oct 2023

PDIG-D-23-00183R1

The relationship between perceptions of electronic health record usability and clinical importance of social and environmental determinants of health on provider documentation

PLOS Digital Health

Dear Dr. Sood,

Thank you for submitting your manuscript to PLOS Digital Health. After careful consideration, we feel that it has merit but does not fully meet PLOS Digital Health's publication criteria as it currently stands. Therefore, we invite you to submit a revised version of the manuscript that addresses the points raised during the review process.

Please submit your revised manuscript within 30 days Nov 29 2023 11:59PM. If you will need more time than this to complete your revisions, please reply to this message or contact the journal office at digitalhealth@plos.org. Please include the following items when submitting your revised manuscript:

We look forward to receiving your revised manuscript.

Kind regards,

Anat Reiner-Benaim

Academic Editor

PLOS Digital Health

Journal Requirements:

Additional Editor Comments (if provided):

Reviewers' comments:

Reviewer's Responses to Questions

**Comments to the Author**

1. If the authors have adequately addressed your comments raised in a previous round of review and you feel that this manuscript is now acceptable for publication, you may indicate that here to bypass the “Comments to the Author” section, enter your conflict of interest statement in the “Confidential to Editor” section, and submit your "Accept" recommendation.

Reviewer #1: All comments have been addressed

Reviewer #3: All comments have been addressed

Reviewer #4: All comments have been addressed

Reviewer #5: All comments have been addressed

2. Does this manuscript meet PLOS Digital Health’s publication criteria? Is the manuscript technically sound, and do the data support the conclusions? The manuscript must describe methodologically and ethically rigorous research with conclusions that are appropriately drawn based on the data presented.

Reviewer #1: Yes

Reviewer #3: Yes

Reviewer #4: Yes

Reviewer #5: Partly

3. Has the statistical analysis been performed appropriately and rigorously?

Reviewer #1: Yes

Reviewer #3: Yes

Reviewer #4: Yes

Reviewer #5: Yes

4. Have the authors made all data underlying the findings in their manuscript fully available (please refer to the Data Availability Statement at the start of the manuscript PDF file)?

Reviewer #1: Yes

Reviewer #3: Yes

Reviewer #4: Yes

Reviewer #5: Yes

5. Is the manuscript presented in an intelligible fashion and written in standard English?

Reviewer #1: Yes

Reviewer #3: Yes

Reviewer #4: Yes

Reviewer #5: Yes

6. Review Comments to the Author

Reviewer #1: Thank you so much to the authors for attending to the comments provided. It is evident in the revisions that attention has been paid to the comments, and subsequently, changes have been made to improve the rigour of the paper.

Reviewer #3: General comments

The authors have addressed all my concerns. The manuscript has greatly improved. 

Minor comment

Sentences on Lines 132-133 should either be deleted or moved to the subtheme: Ethics statement

Reviewer #4: The authors have put in efforts to provide clarity on a number of aspects. Yes the article has greatly improved BUT some few areas are not yet fully clarified. These are few BUT CRITICAL and can be rectified in the shortest time possible by the authors and possibly the article accepted for publication.

SPECIFIC AREAS OF IMPROVEMENT

Add missing information in the abstract as indicated in the abstract comments section

Provide an answer to the so what question asked, include it in the abstract as a critical recommendation. Part of this information is actually in the conclusion section line 366 starting with “improving…”

Add ethical approval number

Provide sampling technique used

Provide the formulae (or in nutshell the calculation) used to arrive at 8% response rate.

Reviewer #5: In author summary section - there should be elaboration on what parts of current infrastructure poses several barriers to accommodate social and environmental determinants of health.

In Introduction section: Please specify the limitations which exist for data capture in existing health frameworks.

7. PLOS authors have the option to publish the peer review history of their article (what does this mean?). If published, this will include your full peer review and any attached files.

**Do you want your identity to be public for this peer review?** For information about this choice, including consent withdrawal, please see our Privacy Policy. 

Reviewer #1: No

Reviewer #3: No

Reviewer #4: No

Reviewer #5: None

---

## [Editor Report · Decision Letter 2]

6 Dec 2023

The relationship between perceptions of electronic health record usability and clinical importance of social and environmental determinants of health on provider documentation

PDIG-D-23-00183R2

Dear Jasani,

We are pleased to inform you that your manuscript 'The relationship between perceptions of electronic health record usability and clinical importance of social and environmental determinants of health on provider documentation' has been provisionally accepted for publication in PLOS Digital Health.

Best regards,

Anat Reiner-Benaim

Academic Editor

PLOS Digital Health